# Reject Illegal Inputs: Scaling Generative Classifiers with Supervised Deep Infomax

## Abstract

Deep Infomax (DIM) is an unsupervised representation learning framework by maximizing the mutual information between the inputs and the outputs of an encoder, while probabilistic constraints are imposed on the outputs. In this paper, we propose Supervised Deep InfoMax (SDIM), which introduces supervised probabilistic constraints to the encoder outputs. The supervised probabilistic constraints are equivalent to a generative classifier on high-level data representations, where class conditional log-likelihoods of samples can be evaluated. Unlike other works building generative classifiers with conditional generative models, SDIMs scale on complex datasets, and can achieve comparable performance with discriminative counterparts. With SDIM, we could perform *classification with rejection*. Instead of always reporting a class label, SDIM only makes predictions when test samples' largest logits surpass some pre-chosen thresholds, otherwise they will be deemed as out of the data distributions, and be rejected. Our experiments show that SDIM with rejection policy can effectively reject illegal inputs including out-of-distribution samples and adversarial examples.

## 1 Introduction

Non-robustness of neural network models emerges as a pressing concern since they are observed to be vulnerable to adversarial examples (Szegedy et al., 2013; Goodfellow et al., 2014). Many attack methods have been developed to find imperceptible perturbations to fool the target classifiers (Moosavi-Dezfooli et al., 2016; Carlini & Wagner, 2017; Brendel et al., 2017). Meanwhile, many defense schemes have also been proposed to improve the robustnesses of the target models (Goodfellow et al., 2014; Tramèr et al., 2017; Madry et al., 2017; Samangouei et al., 2018).

An important fact about these works is that they focus on discriminative classifiers, which directly model the conditional probabilities of labels given samples. Another promising direction, which is almost neglected so far, is to explore robustness of generative classifiers (Ng & Jordan, 2002). A generative classifier explicitly model conditional distributions of inputs given the class labels. During inference, it evaluates all the class conditional likelihoods of the test input, and outputs the class label corresponding to the maximum. Conditional generative models are powerful and natural choices to model the class conditional distributions, but they suffer from two big problems: (1) it is hard to scale generative classifiers on high-dimensional tasks, like natural images classification, with comparable performance to the discriminative counterparts. Though generative classifiers have shown promising results of adversarial robustness, they hardly achieve acceptable classification performance even on CIFAR10 (Li et al., 2018; Schott et al., 2018; Fetaya et al., 2019). (2) The behaviors of likelihood-based generative models can be counter-intuitive and brittle. They may assign surprisingly higher likelihoods to out-of-distribution (OoD) samples (Nalisnick et al., 2018; Choi & Jang, 2018). Fetaya et al. (2019) discuss the issues of likelihood as a metric for density modeling, which may be the reason of non-robust classification, e.g. OoD samples detection.

In this paper, we propose supervised deep infomax (SDIM) by introducing *supervised statistical constraints* into deep infomax (DIM, Hjelm et al. (2018)), an unsupervised learning framework by maximizing the mutual information between representations and data. SDIM is trained by optimizing two objectives: (1) maximizing the mutual information (MI) between the inputs and the high-level data representations from encoder; (2) ensuring that the representations satisfy the supervised statistical constraints. The supervised statistical constraints can be interpreted as a generative

classifier on high-level data representations giving up the *full* generative process. Unlike full generative models making implicit manifold assumptions, the supervised statistical constraints of SDIM serve as explicit enforcement of manifold assumption: data representations (low-dimensional) are trained to form clusters corresponding to their class labels. With SDIM, we could perform classification with rejection (Nalisnick et al., 2019; Geifman & El-Yaniv, 2017). SDIMs reject illegal inputs based on *off-manifold* conjecture (Samangouei et al., 2018; Gu & Rigazio, 2014), where illegal inputs, e.g. adversarial examples, lie far away from the data manifold. Samples whose class conditionals are smaller than the pre-chosen thresholds will be deemed as *off-manifold*, and prediction requests on them will be rejected. The contributions of this paper are :

- We propose Supervised Deep Infomax (SDIM), an end-to-end framework whose probabilistic constraints are equivalent to a generative classifier. SDIMs can achieve comparable classification performance with similar discrinimative counterparts at the cost of small over-parameterization.

- We propose a simple but novel *rejection* policy based on *off-manifold* conjecture: SDIM outputs a class label only if the test sample's largest class conditional surpasses the pre-chosen class threshold, otherwise outputs *rejection*. The choice of thresholds relies only on training set, and takes no additional computations.

- Experiments show that SDIM with rejection policy can effectively reject illegal inputs, including OoD samples and adversarial examples generated by a comprehensive group of adversarial attacks.

## 2    BACKGROUND: DEEP INFOMAX

Deep InfoMax (DIM, Hjelm et al. (2018)) is an unsupervised representation learning framework by maximizing the mutual information (MI) of the inputs and outputs of an encoder. The computation of MI takes only input-output pairs with the deep neural networks based esimator MINE (Belghazi et al., 2018).

Let $E_\phi$ be an encoder parameterized by $\phi$, working on the training set $\mathcal{X} = \{x_i\}_{i=1}^N$, and generating output set $\mathcal{Y} = \{E(x_i)\}_{i=1}^N$. DIM is trained to find the set of parameters $\phi$ such that: (1) the mutual information $\mathcal{I}(X, Y)$ is maximized over sample sets $\mathcal{X}$ and $\mathcal{Y}$. (2) the representations, depending on the potential downstream tasks, match some prior distribution. Denote $\mathbb{J}$ and $\mathbb{M}$ the joint and product of marginals of random variables $X, Y$ respectively. MINE estimates a lower-bound of MI with Donsker-Varadhan (Donsker & Varadhan, 1983) representation of KL-divergence:

$$\mathcal{I}(X, Y) = D_{KL}(\mathbb{J}||\mathbb{M}) \geq \mathbb{E}_{\mathbb{J}}[T_\omega(x, y)] - \log \mathbb{E}_{\mathbb{M}}[e^{T_\omega(x,y)}] \tag{1}$$

where $T_\omega(x, y) \in \mathbb{R}$ is a family of functions with parameters $\omega$ represented by a neural network. Since in representation learning we are more interested in *maximizing* MI, than its exact value, non-KL divergences are also favorable candidates. We can get a family of variational lower-bounds using $f$-divergence representations (Nguyen et al., 2010):

$$\mathcal{I}_f(X, Y) \geq \mathbb{E}_{\mathbb{J}}[T_\omega(x, y)] - \mathbb{E}_{\mathbb{M}}[f^*(T_\omega(x, y))] \tag{2}$$

where $f^*$ is the Fenchel conjugate of a specific divergence $f$. For KL-divergence, $f^*(t) = e^{(t-1)}$. A full $f^*$ list is provided in Tab. 6 of Nowozin et al. (2016). Noise-Contrastive Estimation (Gutmann & Hyvärinen, 2010) can also be used as lower-bound of MI in "infoNCE" (Oord et al., 2018) .

## 3    SUPERVISED DEEP INFOMAX

All the components of SDIM framework are summurized in Fig. 1. The focus of Supervised Deep InfoMax (SDIM) is on introducing supervision to probabilistic constraints of DIM for (generative) classification. We choose to maximize the *local* MI, which has shown to be more effective in classification tasks than maximizing *global* MI (Hjelm et al., 2018). Equivalently, we minimize $\mathcal{J}_{\text{MI}}$:

$$\mathcal{J}_{\text{MI}} = -\frac{1}{M^2} \sum_{i=1}^{M^2} \tilde{\mathcal{I}}(L_\phi^{(i)}(\boldsymbol{x}), E_\phi(\boldsymbol{x})) \tag{3}$$

where $L_\phi(\boldsymbol{x})$ is a local $M \times M$ feature map of $\boldsymbol{x}$ extracted from some intermediate layer of encoder $E$, and $\tilde{\mathcal{I}}$ can be any possible MI lower-bounds.

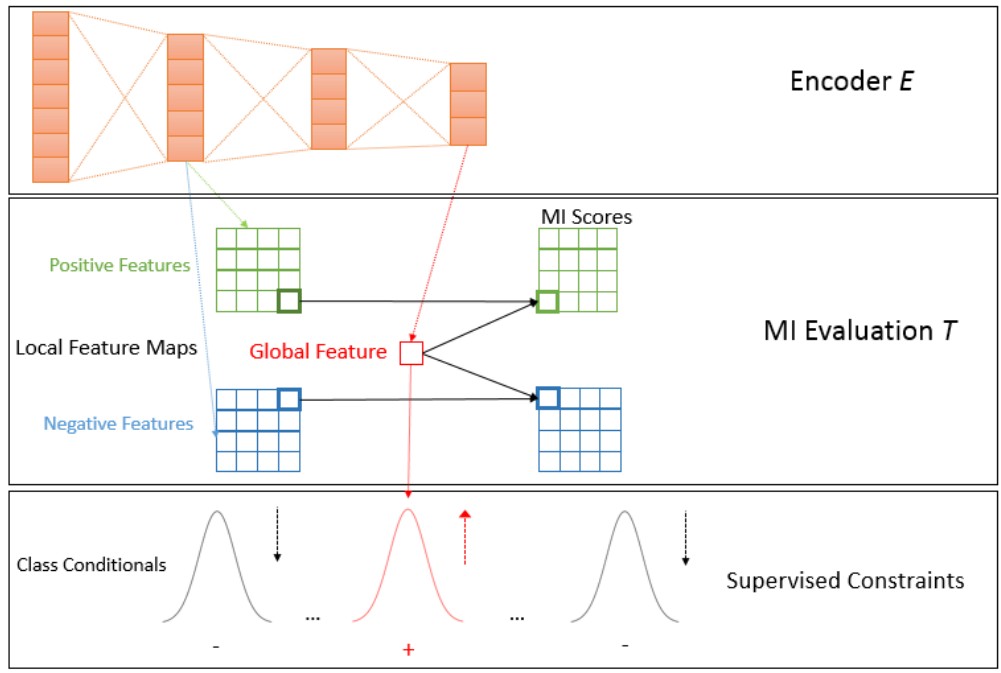

Figure 1: Components of SDIM framework. (1)The encoder $E_\phi$ takes input $\boldsymbol{x}$, and produces pairs of local feature maps $L_\phi(\boldsymbol{x})$ and global representations $E_\phi(\boldsymbol{x})$. (2) The MI evaluation network $T_\omega$ maps every possible positive pairs and negative pairs to MI scores specified by corresponding MI lower-bound. Negative pairs are simply obtained by combine all unpaired local feature maps and global representations within the same mini-batch. (3) Supervised constraints are imposed on the global representations $\tilde{\boldsymbol{x}} = E_\phi(\boldsymbol{x})$ with loss $\mathcal{J}_{\text{NLL}} + \mathcal{J}_{\text{LM}}$ for generative classification. The true class conditionals are maximized, while false class conditionals are minimized. See following parts of this section for details.

## 3.1 Explicit Enforcement of Manifold Assumption

By adopting a generative approach $p(\boldsymbol{x}, y) = p(y)p(\boldsymbol{x}|y)$, we assume that the data follows the *manifold assumption*: the (high-dimensional) data lies on low-dimensional manifolds corresponding to their class labels. Denote $\tilde{\boldsymbol{x}}$ the compact representation generated with encoder $E_\phi(\boldsymbol{x})$. In order to explicitly enforce the manifold assumption, we admit the existence of data manifold in the representation space. Assume that $y$ is a discrete random variable representing class labels, and $p(\tilde{\boldsymbol{x}}|y)$ is the real class conditional distribution of the data manifold given $y$. Let $p_\theta(\tilde{\boldsymbol{x}}|y)$ be the class conditionals we model parameterized with $\theta$. We approximate $p(\tilde{\boldsymbol{x}}|y)$ by minimizing the KL-divergence between $p(\tilde{\boldsymbol{x}}|y)$ and our model $p_\theta(\tilde{\boldsymbol{x}}|y)$, which is given by:

$$D_{KL}\big(p(\tilde{\boldsymbol{x}}|y)||p_\theta(\tilde{\boldsymbol{x}}|y)\big) = \mathbb{E}_{\tilde{\boldsymbol{x}},y\sim p(\tilde{\boldsymbol{x}},y)}[\log p(\tilde{\boldsymbol{x}}|y) - \log p_\theta(\tilde{\boldsymbol{x}}|y)]$$
$$= \mathbb{E}_{\tilde{\boldsymbol{x}},y\sim p(\tilde{\boldsymbol{x}},y)}[\log p(\tilde{\boldsymbol{x}}|y)] - \mathbb{E}_{\tilde{\boldsymbol{x}},y\sim p(\tilde{\boldsymbol{x}},y)}[\log p_\theta(\tilde{\boldsymbol{x}}|y)] \quad (4)$$

where the first item on RHS is a constant independent of the model parameters $\theta$. Eq. 4 equals to maximize the expectation $\mathbb{E}_{\tilde{\boldsymbol{x}},y\sim p(\tilde{\boldsymbol{x}},y)}[\log p_\theta(\tilde{\boldsymbol{x}}|y)]$.

In practice, we minimize the following loss $\mathcal{J}_{\text{NLL}}$, equivalent to empically maximize the above expectation over $\{\tilde{\boldsymbol{x}}_i = E_\phi(\boldsymbol{x}_i), y_i\}_{i=1}^N$:

$$\mathcal{J}_{\text{NLL}} = -\mathbb{E}_{\tilde{\boldsymbol{x}},y\sim p(\tilde{\boldsymbol{x}},y)}[\log p_\theta(\tilde{\boldsymbol{x}}|y)] \approx -\frac{1}{N}\sum_{i=1}^N \log p_\theta(\tilde{\boldsymbol{x}}_i|y_i) \quad (5)$$

Besides the introduction of supervision, SDIM differs from DIM in its way of enforcing the statistical constraints: DIM use adversarial learning (Makhzani et al., 2015) to push the representations to the desired priors, while SDIM directly maximizes the parameterized class conditional probability.

**Maximize Likelihood Margins**   Since a generative classifier, at inference, decides which class a test input $\boldsymbol{x}$ belongs to according to its class conditional probability. On one hand, we maximize samples' true class conditional probabilities (classes they belong to) using $\mathcal{J}_{\text{NLL}}$; On the other hand, we also hope that samples' false class conditional probabilities (classes they do not belong to) can be minimized. This is assured by the following likelihood margin loss $\mathcal{J}_{\text{LM}}$:

$$\mathcal{J}_{\text{LM}} = \frac{1}{N} \cdot \frac{1}{C-1} \sum_{i=1}^{N} \sum_{c=1, c \neq y_i}^{C} \max(\log p(\tilde{\boldsymbol{x}}_i|y=c) + K - \log p(\tilde{\boldsymbol{x}}_i|y=y_i), 0)^2 \quad (6)$$

where $K$ is a positive constant to control the margin. For each encoder output $\tilde{\boldsymbol{x}}_i$, the $C-1$ true-false class conditional gaps are squared[1], which quadratically increases the penalties when the gap becomes large, then are averaged.

Putting all these together, the complete loss function we minimize is:

$$\mathcal{J}_{\text{SDIM}} = \alpha \cdot \mathcal{J}_{\text{MI}} + \beta \cdot \mathcal{J}_{\text{NLL}} + \gamma \cdot \mathcal{J}_{\text{LM}} \quad (7)$$

**Parameterization of Class Conditional Probability**   Each of the class conditional distribution is represented as an isotropic Gaussian. So the generative classifier is simply a embedding layer with $C$ entries, and each entry contains the trainable mean and variance of a Gaussian. This *minimized* parameterization encourages the encoder to learn simple and stable low-dimensional representations that can be easily explained by even unimodal distributions. Considering that we maximize the true class conditional probability, and minimize the false class conditional probability at the same time, we do not choose conditional normalizing flows, since the parameters are shared across class labels, and the training can be very difficult. In Schott et al. (2018), each class conditional probability is represented with a VAE, thus scaling to complex datasets with huge number of classes, e.g. ImageNet, is almost impossible.

### 3.2    Decision Function with Rejection

A generative approach models the class-conditional distributions $p(\boldsymbol{x}|y)$, as well as the class priors $p(y)$. For classification, we compute the posterior probabilities $p(y|\boldsymbol{x})$ through Bayes' rule:

$$p(y|\boldsymbol{x}) = \frac{p(\boldsymbol{x}|y)p(y)}{p(\boldsymbol{x})} \propto p(\boldsymbol{x}|y)p(y)$$

The prior $p(y)$ can be computed from the training set, or we simply use *uniform* class prior for all class labels by default. Then the prediction of test sample $\boldsymbol{x}^*$ from posteriors is:

$$y^* = \arg\max_{c=[1...C]} \log p(\boldsymbol{x}^*|y=c). \quad (8)$$

The drawback of the above decision function is that it always gives a prediction even for illegal inputs. Instead of simply outputting the class label that maximizes class conditional probability of $\boldsymbol{x}^*$, we set a threshold for each class conditional probability, and define our decision function with rejection to be:

$$\begin{cases} y^*, & \text{if } \log p(\boldsymbol{x}^*|y^*) \geq \delta_{y^*} \\ \textit{Rejection}, & \text{otherwise} \end{cases} \quad (9)$$

The model gives a rejection when $\log p(\boldsymbol{x}^*|y^*)$ is smaller than the threshold $\delta_{y^*}$. Note that here we can use $p(\boldsymbol{x}^*|y^*)$ and $p(\tilde{\boldsymbol{x}}^*|y^*)$ interchangeably. This is also known as *selective classification* (Geifman & El-Yaniv, 2017) or *classification with reject option* (Nalisnick et al., 2019)(See Supp. A)

## 4    Related works

**Robustness of Likelihood-based Generative Models**   Though likelihood-based generative models have achieved great success in samples synthesis, the behaviors of their likelihoods can be counter-intuitive. Flow-based models (Kingma & Dhariwal, 2018) and as well as VAEs (Kingma & Welling, 2013), surprisingly assign even higher likelihoods to out-of-distribution samples than the samples in the training set (Nalisnick et al., 2018; Choi & Jang, 2018). Pixel-level statistical analyses in Nalisnick et al. (2018) show that OoD dataset may "sit inside of" the in-distribution dataset (i.e. with roughly the same mean but smaller variance).

---

[1]Using squared margin, we achieve slightly better results in our experiments than simple margin.

***Off-Manifold* Conjecture**   Grosse et al. (2017) observe that adversarial examples are outside the training distribution via statistical testing.

DefenseGAN (Samangouei et al., 2018) models real data distribution with the generator $G$ of GAN. At inference, instead of feeding the test input $x$ to the target classifier directly, it searches for the "closest" sample $G(z^*)$ from generator distribution to $x$ as the final input to the classifier. It ensures that the classifier only make predictions on the data manifold represented by the generator, ruling out the potential adversarial perturbations in $x$. PixelDefend (Song et al., 2017) takes a similar approach which uses likelihood-based generative model - PixelCNN to model the data distribution.

Both DefenseGAN and PixelDefend are additionally trained as peripheral defense schemes agnostic to the target classifiers. Training generative models on complex datasets notoriously takes huge amount of computational resources (Brock et al., 2018). In contrast, the training of SDIM is computationally similar to its discriminative counterpart. The verification of whether inputs are off-manifold is a built-in property of the SDIM generative classifier. The class conditionals of SDIM are modeled on low-dimensional data representations with simple Gaussians, which is much easier, and incurs very small computations.

## 5   EXPERIMENTS

**Datasets**   We evaluate the effectiveness of the rejection policy of SDIM on four image datasets: MNIST, FashionMNIST (both resized to $32 \times 32$ from $28 \times 28$); and CIFAR10, SVHN. See App. B.1 for details of data processing. For out-of-distribution samples detection, we use the dataset pairs on which likelihood-based generative models fail (Nalisnick et al., 2018; Choi & Jang, 2018): FashionMNIST (in)-MNIST (out) and CIFAR10 (in)-SVHN (out). Adversarial examples detection are evaluated on MNIST and CIFAR10.

**Choice of thresholds**   It is natural that choosing thresholds based on what the model knows, i.e. training set, and can reject what the model does not know, i.e. possible illegal inputs. We set one threshold for each class conditional. For each class conditional probability, we choose to evaluate on two different thresholds: *1st* and *2nd* percentiles of class conditional log-likelihoods of the correctly classified training samples. Compared to the detection methods proposed in Li et al. (2018), our choice of thresholds is much simpler, and takes no additional computations.

**Models**   A typical SDIM instance consists of three networks: an encoder, parameterized by $\phi$, which outputs a $d$-dimensional representation; mutual information evaluation networks, i.e. $T_\omega$ in Eqn. (1) and Eqn. (2); and $C$-way class conditional embedding layer, parameterized by $\theta$, with each entry a $2d$-dimensional vector. We set $d = 64$ in all our experiments.

For encoder of SDIM, we use ResNet (He et al., 2016) on $32 \times 32$ with a stack of $8n + 2$ layers, and 4 filter sizes $\{32, 64, 128, 256\}$. The architecture is summarized as:

| output map size | $32 \times 32$ | $16 \times 16$ | $8 \times 8$ | $4 \times 4$ |
|---|---|---|---|---|
| # layers | $1 + 2n$ | $2n$ | $2n$ | $2n$ |
| # filters | 32 | 64 | 128 | 256 |

The last layer of encoder is a $d$-way fully-connected layer. To construct a discriminative counterpart, we simply set the output size of the encoder's last layer to $C$ for classification. We use ResNet10 ($n = 1$) on MNIST, FashionMNIST, and ResNet26 ($n = 3$) on CIFAR10, SVHN.

### 5.1   EVALUATION ON CLEAN DATA

We report the classification accuracies (see Tab. 1) of SDIMs and the discriminative counterparts on clean test sets . Results show that SDIMs achieve the same level of accuracy as the discriminative counterparts with slightly increased number of parameters (17% increase for ResNet10, and 5% increase for ResNet26). We are aware of the existence of better results reported on these datasets using more complex models (Huang et al., 2017; Han et al., 2017) or automatically designed architectures (Cai et al., 2018), but pushing the state-of-the-art is not the focus of this paper.

| Model | # Parameters | MNIST | FashionMNIST | CIFAR10 | SVHN |
|-------|-------------|-------|-------------|---------|------|
| Disc. (ResNet10, $n = 1$) | 1.25M | 99.42% | 94.25% | - | - |
| SDIM (ResNet10, $n = 1$) | 1.46M ( 17% ↑) | 99.55% | 94.58% | - | - |
| Disc. (ResNet26, ($n = 3$)) | 4.39M | - | - | 92.35% | 95.96% |
| SDIM (ResNet26, $n = 3$) | 4.60M ( 5% ↑) | - | - | 92.53% | 95.74% |

Table 1: Clean test accuracies of SDIMs and the discriminative counterparts.

**Is Fully Generative Model Necessary for Generative Classification?** In the evaluations of Li et al. (2018) and Schott et al. (2018), both model class conditional probability with VAE (Kingma & Welling, 2013; Rezende et al., 2014), and achieve acceptable accuracies ($> 98\%$) on MNIST. However, it is hard for *fully* conditional generative models to achieve satisfactory classification accuracies even on CIFAR10. On CIFAR10, methods in Li et al. (2018) achieve only $< 50\%$ accuracy. They also point out that the classification accuracy of a conditional PixelCNN++ (Salimans et al., 2017) is only $72.4\%$. The test accuracy of ABS in (Schott et al., 2018) is only $54\%$. In contrast, SDIM could achieve almost the same performance with similar discriminative classifier by giving up the full generative process, and building generative classifier on high-level representations. Li et al. (2018) improves the accuracy to $92\%$ by feeding the features learned by powerful discriminative classifier-VGG16 (Simonyan & Zisserman, 2014) to their generative classifiers, which also suggests that modeling likelihood on high-level representation (features) is more favorable for generative classification than pixel-level likelihood of *fully* generative classifiers. For classification tasks, discovering discriminative features is much more important than reconstructing the all the image pixels. Thus performing generative classification with full generative models may not be the right choice.

**Decision with Rejection** We also investigate the implications of the proposed decision function with rejection under different thresholds. The results in Tab. 2 show that choosing a higher percentile as threshold will reject more prediction requests. At the same time, the classification accuracies of SDIM on the left test sets become increasingly better. This demonstrate that out rejection policy tend to reject the ones on which SDIMs make wrong predictions.

| Dataset | Original Acc. | $1st$ percentile | | $2nd$ percentile | |
|---------|--------------|-----------------|-----------|-----------------|-----------|
| | | Acc. Left | Rej. Rate | Acc. Left | Rej. Rate |
| MNIST | 99.55% | 99.95% | 3.02% | 99.97% | 4.00% |
| FashionMNIST | 94.58% | 96.45% | 4.63% | 96.94% | 6.60% |
| CIFAR10 | 92.53% | 96.18% | 8.90% | 96.60% | 10.86% |
| SVHN | 95.74% | 97.43% | 3.99% | 98.00% | 6.36% |

Table 2: Classification performances of SDIMs using the proposed decision function with rejection. We report the rejection rates of the test sets and the accuracies on the left test sets for each threshold.

## 5.2 OUT-OF-DISTRIBUTION SAMPLES DETECTION

Class-wise OoD detections are performed, and mean detection rates over all in-distribution classes are reported in Tab. 3. For each in-distribution class $c$, we evaluate the log-likelihoods of the whole OoD dataset. Samples whose log-likelihoods are lower the class threshold $\delta_c$ will be detected as OoD samples. Same evaluations are applied on conditional Glows with *10th* percentile thresholds, but the results are not good. The results are clear and confirm that SDIMs, generative classifiers on high-level representations, are more effective on classification tasks than fully conditional generative models on raw pixels. Note that fully generative models including VAE used in Li et al. (2018); Schott et al. (2018) fail on OoD detection. The stark difference between SDIM and full generative models (flows or VAEs) is that SDIM models samples' likelihoods in the high-level representation spaces, while generative models evaluate directly on the raw pixels. See Supp. C for more results about the histograms of the class conditionals of in-out distributions.

| Model | FashionMNIST(in)-MNIST(out) | CIFAR10(in)-SVHN(out) |
|---|---|---|
| SDIM(*1st* Per.) | 99.36% | 94.24 % |
| SDIM(*2nd* Per.) | 99.64% | 95.81% |
| Glow(*10th* Per.) | 3.53% | 0.02% |

Table 3: Mean detection rates of SDIMs and Glows with different thresholds on OoD detection.

## 5.3 ROBUSTNESS AGAINST ADVERSARIAL EXAMPLES AND DETECTION

We comprehensively evaluate the robustness of SDIMs against various attacks:

- gradient-based attacks: one-step gradient attack FGSM (Goodfellow et al., 2014), its iterative variant projected gradient descent (PGD, Kurakin et al. (2016); Madry et al. (2017)), CW-$L_2$ attack (Carlini & Wagner, 2017), deepfool (Moosavi-Dezfooli et al., 2016).
- score-based attacks: local search attack (Narodytska & Kasiviswanathan, 2016).
- decision-based attack: boundary attack (Brendel et al., 2017).

**Attacks Using Cross-Entropy**   We find that SDIMs are much more robust to gradient-based attacks using cross-entropy, e.g. FGSM and PGD, since the gradients numerically vanish as a side effect of the likelihood margin loss $\mathcal{J}_{\text{LM}}$ of SDIM. This phenomenon is similar to some defences that try to hinder generations of adversarial examples by masking the gradients on inputs. While full generative classifiers in Li et al. (2018) still suffer from these attacks. See Supp. D.1 for detailed results.

**Conservative Adversarial Examples**   Adversarial attacks aim to find the minimal perturbations that sufficiently change the classification labels, i.e. flip other logits to be the largest one. We show case examples on MNIST generated by untargeted attacks and their logits in Tab. 4 (See Supp. D.2 for examples of CIFAR10). Though these attacks successfully flip the logits, they are designed to be conservative to avoid more distortions to the original images. As a result, the largest logits of adversarial examples are still much lower than the thresholds, so they can be detected by our rejection policy. We find that our rejection policy performs perfectly on MNIST, but fails to detect all adversarial examples on CIFAR10 except for Boundary attack (See Tab. 5). It seems to be a well-known observation that models trained on CIFAR10 are more vulnerable than one trained on MNIST. Gilmer et al. (2018) connects this observation to the generalization of models. They found that many test samples, though correctly classified, are close to the misclassfied samples, which implies the existence of adversarial examples. If a model has higher error rate, it would take smaller perturbations to move correctly classified samples to misclassified areas.

| Samples | Original | | DeepFool | | CW-$L_2$ | | Boundary | | LocalSearch | |
|---|---|---|---|---|---|---|---|---|---|---|
| 1st Per. | 477.6 | 465.2 | 481.1 | 477.7 | 465.4 | 407.2 | 474.5 | 470.4 | 472.3 | 463.5 |
| Original | **482.5** | -644.5 | -440.8 | -378.8 | -1082.5 | -409.8 | -473.8 | -850.0 | -415.0 | -699.2 |
| DeepFool | 243.4 | -306.4 | -172.4 | -394.1 | -538.3 | -181.0 | **243.5** | -944.2 | -107.7 | -524.1 |
| CW-$L_2$ | 175.9 | -244.9 | -486.4 | -287.5 | -500.8 | -257.6 | -409.5 | -233.5 | -174.3 | **176.5** |
| Boundary | -58.5 | 11.5 | 205.7 | -149.7 | -415.1 | -308.0 | -356.3 | **205.8** | -223.3 | -250.3 |
| LocalSearch | 180.4 | -225.3 | -481.9 | -281.3 | -498.9 | -223.2 | -378.8 | -257.9 | -143.1 | **189.9** |

Table 4: Full logits of the adversarial examples generated with different attacks. The original image is the first sample of class 0 of MNIST test set. The first row gives the 1st percentile thresholds, and the second row shows the logits of the original image. The largest logits are marked in bold.

**Adversarial examples with more confidence**   Based on the observations above, a natural question we should ask is: can we generate adversarial examples with not only successfully flipped logits,

| Attacks | MNIST | | CIFAR10 | |
|---|---|---|---|---|
| | 1st Per. | 2nd Per. | 1st Per. | 2nd Per. |
| DeepFool | 98.00% | 98.60% | 61.10% | 64.30% |
| Boundary | 100% | 100% | 100% | 100% |
| LocalSearch | 99.90% | 100% | 88.80% | 93.10% |

Table 5: Detection rates of our rejection policies. We perform untargeted adversarial evaluation on the first 1000 images of test sets. CW-$L_2$ is not involved here, but carefully investigated below.

but also the largest logit larger than some threshold value? Unlike the conservativeness on paying more distortions of other attacks, CW attack allows us to control the gap between largest and second largest logits with some confidence value $\kappa$.

We perform targeted CW attacks with confidences $\kappa = \{0, 500, 1000\}$ (Tab. 6). We find that increasing the confidences help increasing the largest logits of adversarial examples to some extent, but may lead to failures of generation. The sensitivity to confidence values is also different given different targets. The success rates of generating adversarial examples monotonically decreases with the confidences increasing (Tab. 7). Note that on discriminative counterparts, CW-$L_2$ with the same settings easily achieves 100% success rates. This means that explicitly forcing data representations to form clusters with maximum margins between them help increase average distances between normal samples and the nearest misclassified areas, thus increase the hardness of finding minimal adversarial perturbations . In this case, it takes a large enough adversarial perturbation to move a sample from its cluster to the other. Meanwhile, detection rates remain satisfactory on MNIST, but obviously decline on CIFAR10. For victim generative classifiers in (Li et al., 2018) under CW-$L_2$ attack, the detection rates of adversarial examples using the proposed detection methods can be $> 95\%$ on MNIST, but fall $< 50\%$ on even CIFAR10-binary (their models don't scale on CIFAR10, and CW-$L_2$ with non-zero confidences are also not evaluated).

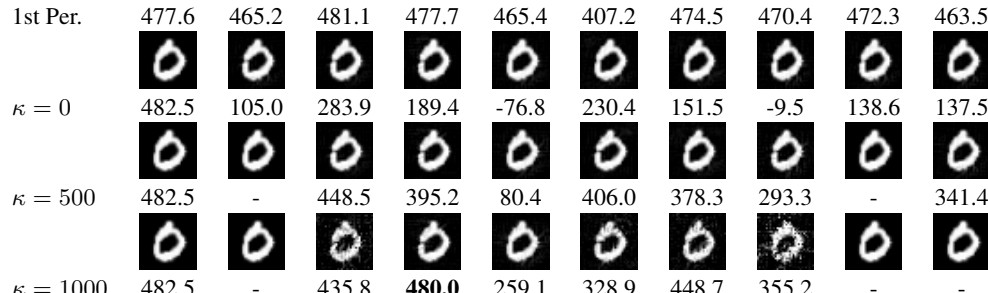

| 1st Per. | 477.6 | 465.2 | 481.1 | 477.7 | 465.4 | 407.2 | 474.5 | 470.4 | 472.3 | 463.5 |
| $\kappa = 0$ | 482.5 | 105.0 | 283.9 | 189.4 | -76.8 | 230.4 | 151.5 | -9.5 | 138.6 | 137.5 |
| $\kappa = 500$ | 482.5 | - | 448.5 | 395.2 | 80.4 | 406.0 | 378.3 | 293.3 | - | 341.4 |
| $\kappa = 1000$ | 482.5 | - | 435.8 | **480.0** | 259.1 | 328.9 | 448.7 | 355.2 | - | - |

Table 6: Adversarial examples generated with targeted CW with different confidences. The original image is the fist sample of class 0. The first row gives the *1st* percentile thresholds. Below the images are the logits corresponding to the given targets. "-" denotes failure of generation.

| Attacks | MNIST | | | CIFAR10 | | |
|---|---|---|---|---|---|---|
| | 1st Per. | 2nd Per. | success rate | 1st Per. | 2nd Per. | success rate |
| CW-$L_2(\kappa = 0)$ | 100% | 100% | 84.65% | 93.93% | 94.80% | 70.84% |
| CW-$L_2(\kappa = 500)$ | 99.78% | 99.78% | 76.61% | 76.55% | 84.07% | 61.26% |
| CW-$L_2(\kappa = 1000)$ | 90.24% | 95.98% | 45.56% | 47.86% | 75.13% | 48.86% |

Table 7: Targeted adversarial evaluations results of our rejection policies on the first 1000 test samples. We report the detection rates with different thresholds and success rates of generating adversarial examples.

**Discussions on off-manifold conjecture**  Gilmer et al. (2018) challenges whether the off-manifold conjecture holds in general. They experiment on synthetic dataset-two high-dimensional concentric

spheres with theoretical analyses, showing that even for a trained classifier with close to zero test error, there may be a constant fraction of the data manifold misclassified, which indicates the existence of adversarial examples *within* the manifold. But there are still several concerns to be addressed: First, as also pointed out by the authors, the manifolds in natural datasets can be quite complex than that of simple synthesized dataset. Fetaya et al. (2019) draws similar conclusion from analyses on synthesized data with particular geometry. So the big concern is whether the conclusions in Gilmer et al. (2018); Fetaya et al. (2019) still hold for the manifolds in natural datasets. A practical obstacle to verify this conclusion is that works modeling the full generative processes are based on manifold assumption, but provide no explicit manifolds for analytical analyses like Gilmer et al. (2018); Fetaya et al. (2019). While SDIM enables explicit and customized manifolds on high-level data representations via probabilistic constraints, thus enables analytical analyses. In this paper, samples of different classes are trained to form isotropic Gaussians corresponding to their classes in representation space (other choices are possible). The relation between the adversarial robustness and the forms and dimensionalities of data manifolds is to be explored. Second, in their experiments, all models evaluated are discriminative classifiers. Considering the recent promising results of generative classifiers against adversarial examples, would using generative classifiers lead to different results? One thing making us feel optimistic is that even though the existence of adversarial examples is inevitable, Gilmer et al. (2018) suggest that adversarial robustness can be improved by minimizing the test errors, which is also supported by our experimental differences on MNIST and CIFAR10.

## 6  CONCLUSIONS

We introduce supervised probabilistic constraints to DIM. Giving up the full generative process, SDIMs are equivalent to generative classifiers on high-level data representations. Unlike full conditional generative models which achieve poor classification performance even on CIFAR10, SDIMs attain comparable performance as the discriminative counterparts on complex datasets. The training of SDIM is also computationally similar to discriminative classifiers, and does not require prohibitive computational resources. Our proposed rejection policy based on *off-manifold* conjecture, a built-in property of SDIM, can effectively reject illegal inputs including OoD samples and adversarial examples. We demonstrate that likelihoods modeled on high-level data representations, rather than raw pixel intensities, are more robust on downstream tasks without the requirement of generating real samples.

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

## A  CLASSIFICATION WITH REJECTION

A very related work to our paper is (Nalisnick et al., 2019), which propose a hybrid model modeling distribution of features $p(\text{features})$ and predictive distribution $p(\text{targets}|\text{features})$ at the same time. Normalizing flow is used to learn invertible features as inputs of discriminative model, i.e. predictive distribution, and provides evaluation of features $\boldsymbol{x}^*$. Inputs out of the training data distribution are rejected by setting a threshold for $p(\boldsymbol{x}^*)$. For SDIM, illagel inputs are rejected by setting thresholds for each of the class conditional. The class conditionals are modeled on the data representations from encoder regularized by MI loss $\mathcal{J}_{\text{MI}}$. The hybrid model in Nalisnick et al. (2019) can successfully distinguish in-distribution dataset and OoD dataset. While SDIM reject illegal inputs including OoD dataset samples and adversarial examples with more fine-grained class conditionals.

Geifman & El-Yaniv (2017) propose a selection method to perform selective classification with desired risk level given a trained model, but they focus on discriminative models.

## B  EVALUATION DETAILS

### B.1  DATA PROCESSING

- MNIST, FashionMNIST: All images are resized to $32 \times 32$ from $28 \times 28$. We scale the images pixels to $[0, 1]$, and normalization is not used for fair comparisons with the baselines in (Li et al., 2018).
- CIFAR10, SVHN: We follow the simple data augmentation in (He et al., 2016) for training: 4 pixels are padded on each side, and a $32 \times 32$ crop is randomly sampled from the padded image or its horizontal flip. For testing, we only evaluate the single view of the original $32 \times 32$ image. All image pixels are also scaled to $[0, 1]$.

No other data augmentations or processings are used except for the explicitly listed above.

### B.2  MODELS AND TRAINING SETTINGS

**MI computation**  The local MI $\mathcal{J}_{\text{MI}}$ is computed between the $4 \times 4$ feature maps and the encoder outputs. We use lower-bound of Jensen-Shannon divergence to estimate the MI $\mathcal{I}_{JSD}(X, Y)$, and leave other lower-bounds unexplored. In practice, we find using other bounds would take more computational resources. However, we think better results can be expected if using these bounds according to the experimental results reported in Hjelm et al. (2018).

**MI evaluation network**  Following Hjelm et al. (2018), we parameterize the MI evaluation network $T_\omega$ as a $1 \times 1$ convolutional neural network with architecture:

| Operation | Size | Activation |
| --- | --- | --- |
| Input $\to 1 \times 1$ conv | 256 | ReLU |
| $1 \times 1$ conv | 256 | ReLU |
| $1 \times 1$ | 1 | |

Table 8: Local MI evaluation concat-and-convolve network architecture.

The input of $T_\omega$ is positive or negative pair of local feature map $L_\phi(\boldsymbol{x})$ and encoder output $E_\phi(\boldsymbol{x})$. The positive pairs, samples from the joint distribution, are constructed by *concat* the encoder output $E_\phi(\boldsymbol{x})$ and the corresponding feature map $L_\phi(\boldsymbol{x})$ of the given batch. The negative pairs, samples from product of marginals, can be constructed by $E_\phi(\boldsymbol{x})$ and shuffled $L_\phi(\boldsymbol{x})$ along the batch-axis.

**Optimization**  We set $\alpha = \beta = \gamma = 1$ in our experiments. The constant $K$ in the likelihood margin loss $\mathcal{J}_{\text{LM}}$ is 10. All models are trained 500 epochs, and we always save the checkpoints reporting the minimum ongoing training losses for evaluation. We use optimizer Adam (Kingma & Ba, 2014) with default learning rate 0.001.

## C    ADDITIONAL RESULTS OF OUT-OF-DISTRIBUTION DETECTION

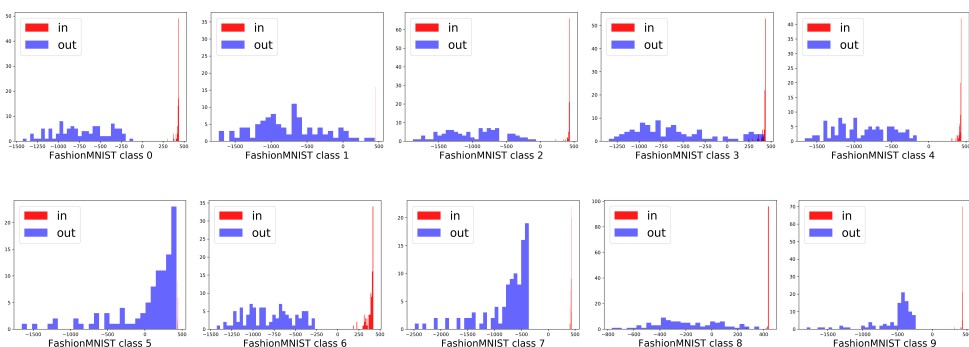

Figure 2: Class-wise distributions of the class conditionals of both in-distribution samples and out-of-distribution samples. The in-out distribution pair is FashionMNIST-MNIST.

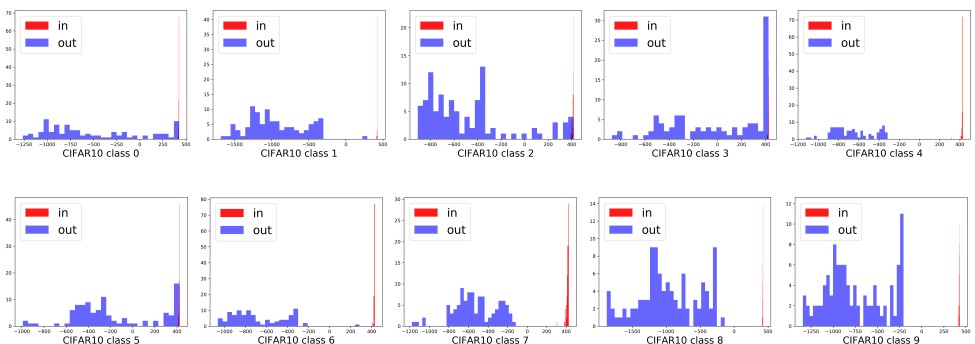

Figure 3: Class-wise distributions of the class conditionals of both in-distribution samples and out-of-distribution samples. The in-out distribution pair is CIFAR10-SVHN.

## D    ADDITIONAL RESULTS OF ADVERSARIAL EVALUATIONS

### D.1    ROBUSTNESS AGAINST ATTACKS USING CROSS-ENTROPY

We make comparisons between SDIM and GBZ (Li et al., 2018), which consistently performs best in *Deep Bayes*.

**FGSM and PGD-$L_\infty$**    The results in Fig 4 and Fig 5 show that SDIM performs consistently better than the baseline. We find that increasing the distortion factor $\epsilon$ of FGSM has no influences of SDIM's accuracy, and the adversarial examples keep the same. Recall that the class conditionals are optimized to keep a considerable margin. Before evaluating the cross entropy loss, softmax is applied on the class conditionals $\log p(\boldsymbol{x}|c)$ to generate a even sharper distribution. So for the samples that are correctly classified, their losses are numerically zeros, and the gradient on inputs $\nabla J_{\boldsymbol{x}}(\boldsymbol{x}, y)$ are also numerically zeros. The PGD-$L_\infty$ we use here is the randomized version (Madry et al., 2017)[2], which adds a small random perturbation before the iterative loop. The randomness is originally introduced to generate different adversarial examples for adversarial training, but here it breaks the zero loss so that the gradient on inputs $\nabla J_{\boldsymbol{x}}(\boldsymbol{x}, y)$ will not be zeros in the loop. FGSM can also be randomized (Tramèr et al., 2017), which can be seen as a one-step variant of randomized PGD.

---

[2]We use cleverhans in this evaluation. There are two types of implementations in cleverhans. By default *rand_init* is set to True, the PGD is randomized. If *rand_init* is False, then the implementation is Basic Iterative Method (Kurakin et al., 2016)

This phenomena is similar to what some defenses using gradient obfuscation want to achieve. Defensive distillation (Carlini & Wagner, 2016) masks the gradients of cross-entropy by increasing the temperature of softmax. But for CW attacks, which do not use cross-entropy, and operate on logits directly, this could be ineffective.

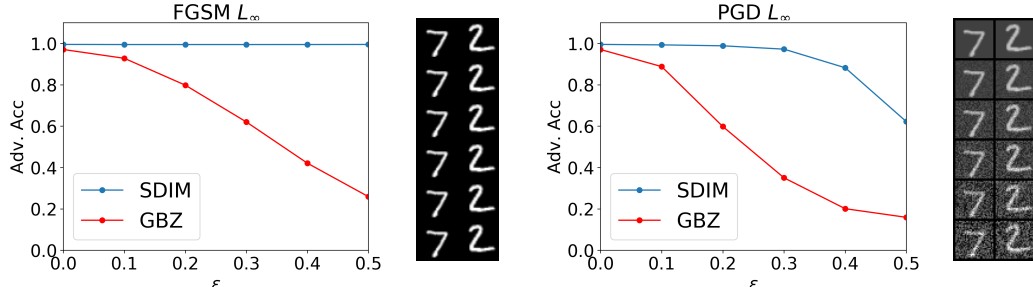

Figure 4: The adversarial classification accuracies of SDIM and GBZ on MNIST under FGSM-$L_\infty$ and PGD-$L_\infty$ attacks. On the right are the generated adversarial examples with $\epsilon$ from 0 to 0.5.

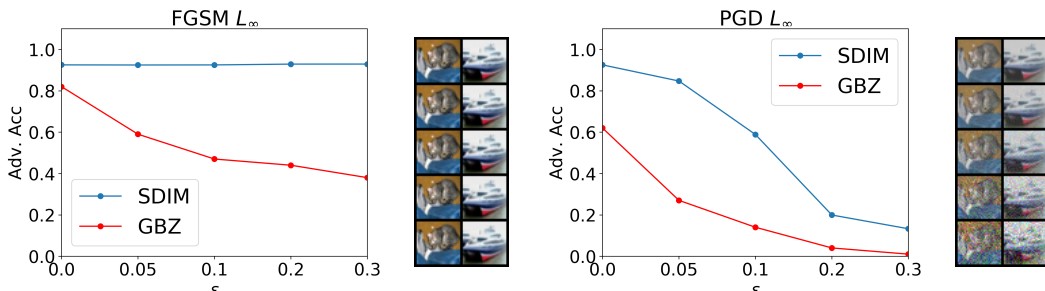

Figure 5: The adversarial classification accuracies of SDIM and GBZ on CIFAR10 under FGSM-$L_\infty$ and PGD-$L_\infty$ attacks. On the right are the generated adversarial examples with $\epsilon$ from 0 to 0.5.

## D.2 ADVERSARIAL EXAMPLES OF CIFAR10

| Samples | original | | deepfool | | cw-$L_2$ | | boundary | | JSMA | |
|---|---|---|---|---|---|---|---|---|---|---|
| 1st Per. | 408.6 | 396.6 | 375.9 | 378.4 | 363.3 | 376.7 | 409.4 | 383.0 | 397.1 | 412.2 |
| original | **424.2** | -386.1 | -379.2 | -319.8 | -370.2 | -357.2 | -356.9 | -259.3 | -291.9 | -239.0 |
| DeepFool | 153.4 | -391.6 | -344.1 | -262.6 | -376.1 | -345.9 | **215.1** | -306.5 | -244.3 | -326.4 |
| CW-$L_2$ | 129.9 | -555.3 | **235.4** | -353.1 | -471.6 | -400.3 | -342.7 | -367.2 | -486.4 | -326.4 |
| Boundary | 213.9 | -417.4 | -458.3 | -548.0 | -587.4 | -236.3 | **214.0** | -1246.1 | -171.2 | -555.6 |
| LocalSearch | 165.2 | -485.7 | **190.9** | -325.6 | -439.0 | -379.0 | -318.8 | -327.5 | -357.9 | -272.3 |

Table 9: Full logits of adversarial examples generated with different attacks. The original image is the fist sample of class 0 of CIFAR10 test set. The first row gives the 1st percentile thresholds, and the second row shows the logits of the original image. The largest logits are marked in bold.

