# OpenReview forum: "Reject Illegal Inputs: Scaling Generative Classifiers with Supervised Deep Infomax"
_ICLR.cc/2020/Conference — Reject_

### Official Review · AnonReviewer1 · 2019-10-23
**Official Blind Review #1**

**Rating:** 3

**Review:**

The paper proposed a supervised method for robust classification. By utilizing mutual information constrain on the encoder, supervised probabilistic constraint on the class conditional probability, and introducing a margin to the maximum likelihood, the proposed method can manage a high classification accuracy and detect the out of distribution data.

The motivation is good, the writing is OK. The structure of the paper needs to be refined.  The experiments are not very strong. Several concerns are listed:

1.	It will be better if the author can show a structure map for Encoder, classification, class condition embedding, and mutual information evaluation networks, to clarify their relationships.
2.	In Table 2, what is the accuracy for the rejection (outlier detection)?
3.	The choose of the threshold for rejection is tricky. It will be better if the author can provide some rules for choosing the threshold which can be generalized to a new dataset.
4.	The overall loss functions have three components, what is the contribution of different components to the final performance experimentally?
5.	The architecture for encoder is large. Is it proper for Mnist which is a relatively simple dataset?
6.	The comparison talked in the paragraph “Is Fully Generative Model Necessary for Generative Classification”, are these accuracies obtained from a comparable network size? It only makes sense if they are obtained using a comparable parameter size.


**Experience Assessment:**

I have published one or two papers in this area.

**Review Assessment: Checking Correctness Of Derivations And Theory:**

I assessed the sensibility of the derivations and theory.

**Review Assessment: Checking Correctness Of Experiments:**

I assessed the sensibility of the experiments.

**Review Assessment: Thoroughness In Paper Reading:**

I read the paper at least twice and used my best judgement in assessing the paper.

---

> ### Author Response · Authors · 2019-11-11
> **Responses**
>
> Thank you for your comments.
>
> 1.  Your suggestion of a structure map was exactly what we wanted to do, but failed due to the 8-page submission limit. We've added a structure map of SDIM framework in the revised paper. Please check.
>
> 2.  Table 2 is the classification accuracies on the clean test datasets.  Since the thresholds we choose are 1or2 percentiles of training samples' class conditionals, a small portion of "legal" (clean) test samples may be rejected. So we report the rejection rates of "legal" samples, as well as the classification accuracies on the left test samples. We can see, if we choose a higher threshold, i.e. 2-th percentile, more "legal" requests will be rejected by SDIM without a prediction label; but SDIM becomes more confident and accurate on the left test sets.
>
> 3. The choice of the threshold is actually very simple. We choose one threshold for each class only based on the training sets. Say for class $c$, the class conditionals of all training samples is a set $S_c = \{y_i\}_{i=1}^{N_c}$. 1-th percentile is simply the (0.01 * $N_c$) th smallest value of set $S_c$, so on and so forth.  Note that this also means 1 percent of the training samples will be rejected without outputing prediction label by our rejection decision function.
>
> 4. In our experiments, we use  MI lower-bound  specified by Jensen-Shannon (JS) divergence, our MI losses are very close to, also bounded by, the maximum of JS divergence $\log 2$ . The likelihood margin loss $\mathcal{J}_{\text{LM}}$ usually reduce to zero.   With all these two bounded, the NLL is minimized, but is unbounded.
>
> 5. I understand that even a trivial CNN model can easily get >99% accuracy on MNIST. In our experiments, for clarity and consistency, we use a popular definition of residual networks that can be universally evaluated on CIFAR10, SVHN, FashionMNIST, MNIST. For MNIST and FashionMNIST, we simply choose the simplest network of the definition($n=1$ for each residual stage),  and no special attention is paid.  It would also not surprising that  achieving the same-level results with much smaller network.
>
> 6. I understand your concern about the fairness of comparisons. But for here, discussing about the network size is actually meaningless, and also not applicable:  (1) Poor performance. On CIFAR10, [1] reports <50% accuracy, and [2] is 54%. (2) Networks sizes of [1] [2] not available. [1] simply mentioned the poor result (<50%) on CIFAR10 in their preliminary experiments (the network structure not available in the open-sourced code); their generative classifier in the paper is trained on the features of VGG16, which is not comparable with SDIM. [2] also simply mentioned the poor result on CIFAR10, and their models are completely evaluated  on MNIST.
>
> [1] Li, Yingzhen et al. Are generative classifiers more robust to adversarial attacks? ICML 2019
>
> [2] Lukas Schott, et al. Towards the first adversarially robust neural network model on mnist. arXiv preprint arXiv:1805.09190, 2018

---

### Official Review · AnonReviewer2 · 2019-10-25
**Official Blind Review #2**

**Rating:** 8

**Review:**

This paper studies classification problems via a reject option. A reject option could be useful in prediction problems to handle Out-of-distribution examples. The classification procedure studied in this paper builds on three components 1. An auto-encoder that obtains a latent low-dimensional representation of the data point 2.  A generative model that models the class-conditional probability model and 3.  a margin based loss function that learns a classifier that provides a large probability mass to the class-conditional distribution corresponding to the correct class.  The final decision procedure is to reject an input if the best class conditional probability is small and to use the class corresponding to the best class conditional probability otherwise.

On the whole I like the paper and think that the problem tackles an important problem. I have a few comments
1. I would like to see what is the log-likelihood assigned by the proposed procedure on OOD samples and would like to see a comparison of the log-likelihood assigned by other procedures.

**Experience Assessment:**

I do not know much about this area.

**Review Assessment: Checking Correctness Of Derivations And Theory:**

N/A

**Review Assessment: Checking Correctness Of Experiments:**

I assessed the sensibility of the experiments.

**Review Assessment: Thoroughness In Paper Reading:**

I read the paper at least twice and used my best judgement in assessing the paper.

---

> ### Author Response · Authors · 2019-11-11
> **Responses**
>
> Thank you for your comments.
>
> We've added more results of OOD detection (histograms of in-out samples' assigned class conditionals) in Section C of Supplementary part.  Please Check.

---

### Official Review · AnonReviewer4 · 2019-11-03
**Official Blind Review #4**

**Rating:** 3

**Review:**

The paper proposes a scalable approach to train generative classifiers using information maximizing representation learning, with the motivation that generative classifiers could be more robust to adversarial attacks than discriminative classifiers. An off-the-shelf mutual information maximizer (MINE, DIM) is used to learn low-dimensional representations of images. Then, class-conditioned generative models of the representations are learned avoiding full generative modeling of the images. An additional loss is used to train the generative classifier which maximizes likelihood margins. Finally, percentile-based thresholds of the class log-probabilities is proposed to be used to reject classification for out-of-manifold inputs.

The paper cites existing literature which indicates that generative classifiers might be more robust to adversarial attacks, and uses recently-proposed representation learning techniques to scale up learning generative generative classifiers. The motivation of the proposed technique is clear, and the problem itself is relevant. Similar prior work use generative modeling at the pixel-level. Generative modeling of representations is novel, afaik.

The technique is evaluated on out-of-distribution sample detection (FashionMnist->MNIST and Cifar10->SVHN) and adversarial attacks. FGSM, PGD, CW-L2 attack, deepfool.

The experiments section is not very clearly written. Some of the evaluation itself is nonstandard in which only the first example of digit 0 of MNIST is used. The paper needs to have a clearer explanation and interpretation of the results. It’s not clear what the logits in table 4 and table 6 are, and what is being shown by the comparison.

Summary: The authors propose a new technique for training generative classifiers with the aim to improve robustness to adversarial attacks and confidence on out-of-distribution samples. The method is well-motivated and explained, but the experiment section is not very clearly written and I’m not confident whether the technique represents an advancement in the state-of-the-art or not.

Misc comments:

“By adopting a generative approach p(x, y) = p(y)p(x|y), we assume that the data follows the manifold assumption: the (high-dimensional) data lies on low-dimensional manifolds corresponding to their class labels.”

This sentence seems to be making a stronger claim than is needed, which is possibly incorrect. Assuming a generative approach doesn’t require assuming the manifold assumption.

Nit: “possible MI low-bounds”
Nit: “state-of-the-arts”
Typo: “The original image is the fist sample of class 0”


**Experience Assessment:**

I have read many papers in this area.

**Review Assessment: Checking Correctness Of Derivations And Theory:**

N/A

**Review Assessment: Checking Correctness Of Experiments:**

I did not assess the experiments.

**Review Assessment: Thoroughness In Paper Reading:**

I read the paper at least twice and used my best judgement in assessing the paper.

---

> ### Author Response · Authors · 2019-11-12
> **Responses**
>
> Thank you for your comments.
>
> About manifold assumption:
>
> Manifold assumption is a very basic assumption  \emph{implicitly}  made by many ideas behind machine learning (See [1] section 5.11.3 Page 161-164 of Ian Goodfellow's Deep Learning book). In particular, one important citation of this paper, [2] make manifold assumption very clear in its section 2.
>
> The key difference of our paper is that:  (1) we enforce the manifold on the representations learned from data; (2) the form of the manifold  is explicitly constrained (i.e. supervised constraints of SDIM) to perform generative classification. This also constitutes the foundation of our rejection decision function. Illegal inputs (out-of-distribution samples or adversarial examples) will be rejected if their class conditionals are smaller than the corresponding thresholds.
>
> About Experiments:
>
> The logits in Table 4 and Table 6 are simply class conditionals.  Table 4 and 6 are provided as typical cases that how the adversarial examples are rejected (detected) by the chosen thresholds (listed are 1st percentile thresholds). In table 4, inspecting the logits, i.e. class conditionals of adversarial examples. If we simply output the label corresponding to the maximum (like what typical discriminative classifiers do), we make a wrong prediction. But we use the rejection decision function, we will successfully reject (detect) them (highest logits smaller than thresholds).
>
> I'd like to provide some clarification about the whole experiments.
>
> In short:
>
> (1) show that SDIMs are able to achieve same-level accuracies as discriminative counterparts (previous generative classifiers based on full generative models report very poor accuracies (<80%) on CIFAR10)
>
> (2) effectively reject illegal inputs (OoD samples or adversarial examples) based on off-manifold conjecture, i.e. reject samples if their class conditionals are smaller than thresholds.
>
> Typos are corrected in the updated revision, please check.
>
> We would like to address your further concerns about this paper if you have any.
>
>
> [1] Deep Learning, Ian Goodfellow and Yoshua Bengio and Aaron Courville.
>
> [2] Li, Yingzhen et al. Are generative classifiers more robust to adversarial attacks? ICML 2019

---

### Decision · Program_Chairs · 2019-12-19

**Decision:**

Reject

**Comment:**

This paper combines a well-known, recently proposed unsupervised representation learning technique technique with a class-conditional negative log likelihood and a squared hinge loss on the class-wise conditional likelihoods, and proposes to use the resulting conditional density model for generative classification. The empirical work appears to validate the claim that their method leads to good out of distribution detection, and better performance using a rejection option. The adversarial defense results are less clear. Reporting raw logits is a strange choice, and difficult to interpret; the table is also difficult to read, and this method of reporting makes it difficult to compare against existing methods.

The reviewers generally remarked on presentation issues. R1 asked about the contribution of various loss terms, a matter I feel is underexplored in this work, and the authors mainly replied with a qualitative description of loss behaviour in the joint system, which I don't believe was the question. R1 also asked about the choice of thresholds and the issues of fairness of comparison regarding model capacity, neither of which seemed adequately addressed. R3 remarked on the clarity being lacking, and also that "Generative modeling of representations is novel, afaik." (It is not; see, for example, the VQ-VAE line of work where PixelCNN priors are fit on top of representations, and layer-wise pre-training works of the mid 2000s, where generative models were frequently fit on greedily trained feature representations, sometimes in conjunction with a joint generative model of class labels).  R2's review was very brief, and with a self-reported low confidence, but their concerns were addressed in a subsequent update.

There are three weaknesses which are my grounds for recommending rejection. First, this paper does a poor job of situating itself in the wider body of literature on classification with rejection, which dates to at least the 1970s (see Bartlett & Wengkamp, 2006 and the references therein). Second, the empirical work makes little comparison to other methods in the literature; baselines on clean data are self-generated, and the paper compares to no other adversarial defense proposals. In a minor drawback, ImageNet results are also missing; given that one of the purported advantages of the method is scalability, a large scale benchmark would have strengthened this claim. Third, no ablation study is undertaken that might give us insight into the role of each term of the loss. Given that this is a straightforward combination of well-understood techniques, a fully empirical paper ought to deliver more insight into the combination than this manuscript has.

---

> ### Author Response · Authors · 2019-12-21
> **Thank you for your detailed and helpful meta-review**
>
> Thank you for your detailed and helpful meta-review. We are so grateful for this, and will improve this draft accordingly.
>
> The only sad thing is that we can only see these points in meta-review, and fail to receive any further responses in the rebuttal phase. So that we have no chance to address them in the rebuttal, and only to see this paper rejected.